# Federated Incomplete Multi-view Clustering with Globally Fused Graph Guidance

**Guoqing Chao\*** [1]  **Zhenghao Zhang** [1]  **Lei Meng** [2]  **Jie Wen** [3]  **Dianhui Chu** [1]

## Abstract

Federated multi-view clustering has been proposed to mine the valuable information within multi-view data distributed across different devices and has achieved impressive results while preserving the privacy. Despite great progress, most federated multi-view clustering methods only used global pseudo-labels to guide the downstream clustering process and failed to exploit the global information when extracting features. In addition, missing data problem in federated multi-view clustering task is less explored. To address these problems, we propose a novel Federated Incomplete Multi-view Clustering method with globally Fused Graph guidance (FIMCFG). Specifically, we designed a dual-head graph convolutional encoder at each client to extract two kinds of underlying features containing global and view-specific information. Subsequently, under the guidance of the fused graph, the two underlying features are fused into high-level features, based on which clustering is conducted under the supervision of pseudo-labeling. Finally, the high-level features are uploaded to the server to refine the graph fusion and pseudo-labeling computation. Extensive experimental results demonstrate the effectiveness and superiority of FIMCFG. Our code is publicly available at https://github.com/PaddiHunter/FIMCFG.

## 1. Introduction

With the fast development of information collection techniques, data can be obtained from different views, resulting in multi-view data (Chao et al., 2025a). Multi-view clustering (MVC) is a popular machine learning paradigm designed to group data using complementary and consistent information from multiple views. In recent years, deep learning has been widely used in MVC, known as deep multi-view clustering, which has achieved state-of-the-art clustering performance due to its excellent representation learning ability (Chen et al., 2022; Yan et al., 2021; Yu et al., 2024). In practice, due to the complexity of the data collection and transmission process, data may be missing for some views, leading to the Incomplete Multi-view Problem (IMP) (Lin et al., 2021). To address IMP, some incomplete multi-view clustering methods adopt data recovery methods to predict missing data. Recently, graph neural networks have received much attention for their ability to capture structural information. In incomplete multi-view clustering field, graph convolutional neural networks (GCNs) can compute missing data features with the help of graph structural information and neighboring node attributes, reducing the impact of missing data (Chao et al., 2024).

Federated learning is a distributed machine learning paradigm that allows multiple clients to jointly train an ensemble model without exposing raw data information (Liu et al., 2020; Qi et al., 2025b). It aims for dual optimization objectives of global generalization and local personalization(Meng et al., 2024), and focuses on addressing the challenges posed by heterogeneous data distributions and missing classes(Qi et al., 2023; 2025a). Federated multi-view learning is designed to learn a global model from multi-view data distributed across different clients. The main challenges are formulated from privacy preservation and the heterogeneity of multi-view data. In addition, multi-view data distributed on different clients is often incomplete and some views may be missing, exacerbating the complexity (Chen et al., 2023).

There exist some research works combining graph neural networks (GNN) with federated multi-view clustering (Yan et al., 2024). However, most of current GNN-based federated multi-view learning only used the graph structure information accompanied by noise and didn't tackle the missing data problem. In addition, in most federated multi-view clustering models, the client tends to extract features for a single view and on the server, the features are fused

[1]School of Computer Science and Technology, Harbin Institute of Technology, Weihai, China [2]School of Software, Shandong University, Jinan, China [3]Shenzhen Key Laboratory of Visual Object Detection and Recognition, Harbin Institute of Technology, Shenzhen, China. Correspondence to: Guoqing Chao <guoqingchao@hit.edu.cn>.

*Proceedings of the 42$^{nd}$ International Conference on Machine Learning*, Vancouver, Canada. PMLR 267, 2025. Copyright 2025 by the author(s).

to compute pseudo-labels acting as global self-supervised information to guide the client training. In summary, most of these approaches only mine the global information for downstream clustering, while ignoring the interaction of multiple views in upstream feature extraction, making it difficult to capture the global information within multiple views, degrading the clustering performance.

To solve the above problems, we propose the Federated Incomplete Multi-view Clustering method with globally Fused Graph guidance (FIMCFG). Clients include upstream two-level feature extraction and downstream clustering. To address incomplete data problem, the global graph structure migration is proposed, which repairs the incomplete local graph. With the information propagation mechanism of GCN, the latent feature of missing data can be learned from the neighboring nodes in repaired graph. With the guidance of globally fused graph, two-level feature extraction has three parts: dual-head graph encoder, decoder and fusion module. We input the fused graph from the server and the repaired local graph into the dual-head graph encoder to extract underlying features including global and view specific information. Due to rich view specific information included, underlying features are used by decoder to reconstruct data and a fusion module is introduced to fuse the underlying features into a low-dimensional high-level feature under the guidance of globally fused graph. With the fused graph, global information is propagated in feature extraction. Then, we perform clustering on the high-level features and optimize the clustering layer with the global supervision of pseudo-label. At the server, we perform graph fusion, feature fusion and global clustering to obtain the fused graphs and pseudo-labels, which are used as global information to guide the client. The main contributions of our work are summarized as follows:

- A federated incomplete multi-view clustering framework with dual-head GCN encoders is proposed. By introducing the globally fused graph guidance, the encoders on clients are able to to grasp the global information among views.

- The global graph structure migration is proposed to repair incomplete local graphs, which is used to estimate latent features of missing data. It improves the accuracy of the estimated features of missing data. Extensive experiments on real-world multi-view data sets demonstrate the superior performance of our proposed model.

## 2. Related Works

### 2.1. Incomplete Multi-View Clustering

In real-world applications, multi-view data often suffer from missing data problems due to uncontrollable data collec-

tion, transmission, or storage factors. Incomplete Multi-View Clustering (IMVC) addresses this challenge by learning the robust clustering structures from partially observed multi-view data. Recent advances in deep learning have significantly enhanced IMVC performance, with various innovative approaches proposed. For instance, Completer (Lin et al., 2021) leverages the autoencoders to maximize the cross-view mutual information via contrastive learning, ensuring view-consistent representations. Additionally, it employs the dual prediction to minimize the conditional entropy, effectively recovering the missing views. Another approach, based on variational autoencoders(Xu et al., 2024), utilizes the Product-of-Experts (PoE) method to aggregate multi-view information, deriving a shared latent representation to handle incompleteness. Further improving data recovery, AIMC(Xu et al., 2019) integrates the element-wise reconstruction with Generative Adversarial Networks (GANs) to generate the plausible missing data. More recently, Graph Neural Networks (GNNs) have been introduced to IMVC, capitalizing on their ability to model relational data. For example, ICMVC(Chao et al., 2024) tackles the missing data through multi-view consistency relation transfer combined with Graph Convolutional Networks (GCNs). Similarly, CRTC(Wang et al., 2022) introduces a cross-view relation transfer completion module, where GNNs infer missing data based on transferred relational graphs.

### 2.2. GNN based Multi-View Clustering

In recent years, GNNs have been widely used in multi-view clustering due to their powerful feature extraction ability to exploit the graph structure and node attribute information, and have received more and more attentions (Xia et al., 2022; Du et al., 2023). GNNs have also been studied to transfer the inter-view graph structure to deal with the missing data problem. For instance, Chao et al. (2024) proposed to use multi-view consistency relation migration and GCN to tackle the missing data problem in multi-view clustering task. In addition, several studies have found that propagating information within the single view could limit the performance. To solve this problem, Xiao et al. (2023) proposed a dual fusion module and a dual information propagation mechanism to capture multiple information of different views. Wang et al. (2024b) learned the consensus representations through the unified heterogeneous attribute graphs, which can propagate structural information across multiple views to improve the feature representation. Chao et al. (2025b) proposed a hierarchical information-transfer incomplete multi-view clustering method that integrates view-specific representation learning, global graph propagation, and contrastive clustering.

However, the aforementioned GNN-based MVC methods can only address the missing data problem and informa-

tion propagation limitations in centralized environments. Although some federated learning multi-view clustering methods have been proposed for distributed scenarios, their clients often do not explicitly consider the global information in the upstream feature extraction process. In addition, there are fewer studies on how to exploit the multi-view clustering capability of GNNs in distributed environments. In this paper, we propose a novel GCN-based federated multi-view clustering (FedMVC) approach, which solves the missing data problem and restricted global information propagation in distributed scenarios.

## 2.3. Federated Multi-View Clustering

Federated Multi-View Clustering is an emerging task which aims to conduct clustering task for multi-view data through collaboration of clients. The heterogeneity of multi-view data and the importance of privacy preservation make it extremely difficult to handle. Chen et al. (2023) proposed sample alignment and data extension techniques to explore the complementary cluster structures of multiple views. Based on Chen et al. (2023), Ren et al. (2024) utilized sample commonality and view generality to adaptively generate alignment matrices to further address data misalignment across view clients. To deal with the heterogeneous hybrid views problem, Chen et al. (2024) designed a local collaborative contrastive learning approach to address the client gaps and a global specific weighted aggregation method to reduce view gaps. Yan et al. (2024) adopted the heterogeneous GNN encoders to address the data heterogeneity at clients and designed a global pseudo-labeling mechanism with heterogeneous aggregation in a federated environment to deal with the incomplete view problem.

Although federated multi-view clustering works have made great progress, they tend to extract the features at the client first, and then aggregate the features at the server and compute the pseudo-labels to transfer to the client as self-supervised information. Pseudo-labeling often only guides the downstream clustering process and is not useful for the upstream feature extraction process, which limits the feature extraction process from utilizing global information. To solve this problem, we propose FIMCFG to improve the feature representation of clients by effectively using the fused graph from the server, such that the dual-head graph encoder grasps the global information of multiple views.

# 3. Proposed Method

## 3.1. Problem Formulation

We give a formal definition of federated multi-view clustering. Suppose there is a dataset containing $N$ samples with $M$ views distributed over $M$ clients (denoted as $X = \{X^1, X^2, ..., X^M\}$), which will be divided into $K$ clusters. Each client has only one view-specific data $X^m = \{x_1^m, x_2^m, ..., x_{N_m}^m\} \in \mathbb{R}^{N_m \times D_m}$, where $D_m$ denotes the dimensionality of the samples in view $m$. Due to incomplete samples in the clients, $N_m \leq N$. In order to facilitate the processing, for those missing samples, we use the zero vector $\mathbf{0}$ to fill in, i.e., $X^m \in \mathbb{R}^{N \times D_m}$. The graph structure of the raw data is represented by adjacency matrices $A = \{A^1, A^2, ..., A^M\}$, and $A^m \in \{0, 1\}^{N \times N}$. $A_{ij}^m = 1$ or $0$ denotes the presence or absence of edges between $x_i^m$ and $x_j^m$. The adjacency matrix of the fused graph from the server is denoted as $\overline{A} \in \{0, 1\}^{N \times N}$.

Our model architecture consists of $M$ clients and one server. Each client utilizes its private data for local view training. The high-level features $\{H^m\}_{m=1}^M$ are obtained through two level feature extraction guided by the globally fused graph $\overline{A}$. Then clustering is performed based on them. Based on the clustering results, we compute the weights $\{w^m\}_{m=1}^M$ by high-level feature. After that, the server receives $\{H^m\}_{m=1}^M$ and $\{w^m\}_{m=1}^M$ from the clients and performs feature fusion, graph fusion and global clustering. The overview illustration of the model is shown in Figure 1.

## 3.2. Client Local Training with Global Guidance

As shown in Figure 1, The client contains two steps with feature extraction and clustering. client $M$ extracts the underlying features $\widetilde{H}^M$ and high-level features $H^M$ under the guidance of fused graph $\overline{A}$, such that they can capture the global information of multiple views. Through the initialization of the global clustering center $U^M$ and the supervised training with pseudo-labels $P$, the clustering layers of different clients are aligned and obtain a consistent clustering structure. Subsequently, we take client $m$ as an example to introduce the local training of clients.

### 3.2.1. GLOBAL GRAPH STRUCTURE MIGRATION

In the beginning, because there is no graph structure information on client $m$, the local graph adjacency matrix $A^m \in \{0, 1\}^{N \times N}$ needs to be constructed from the raw data. We use the radial basis function to compute the similarity matrix $S^m \in [0, 1]^{N \times N}$, which is calculated as follows:

$$S_{ij}^m = e^{-\frac{\|x_i^m - x_j^m\|_2^2}{t}}, \tag{1}$$

where $S_{ij}^m \in [0, 1]$ denotes the similarity between $x_i^m$ and $x_j^m$. After that, we set the largest $k$ elements of each row to 1 and the others to 0 in $S^m$ to construct the adjacency matrix $A^m$, where 1 or 0 denotes the presence or absence of edges.

To handle the incomplete data problem, global graph structure migration is proposed. Under the GCN encoder on clients, each sample estimates single-view features according to its attribute and neighboring nodes' attributes. Since

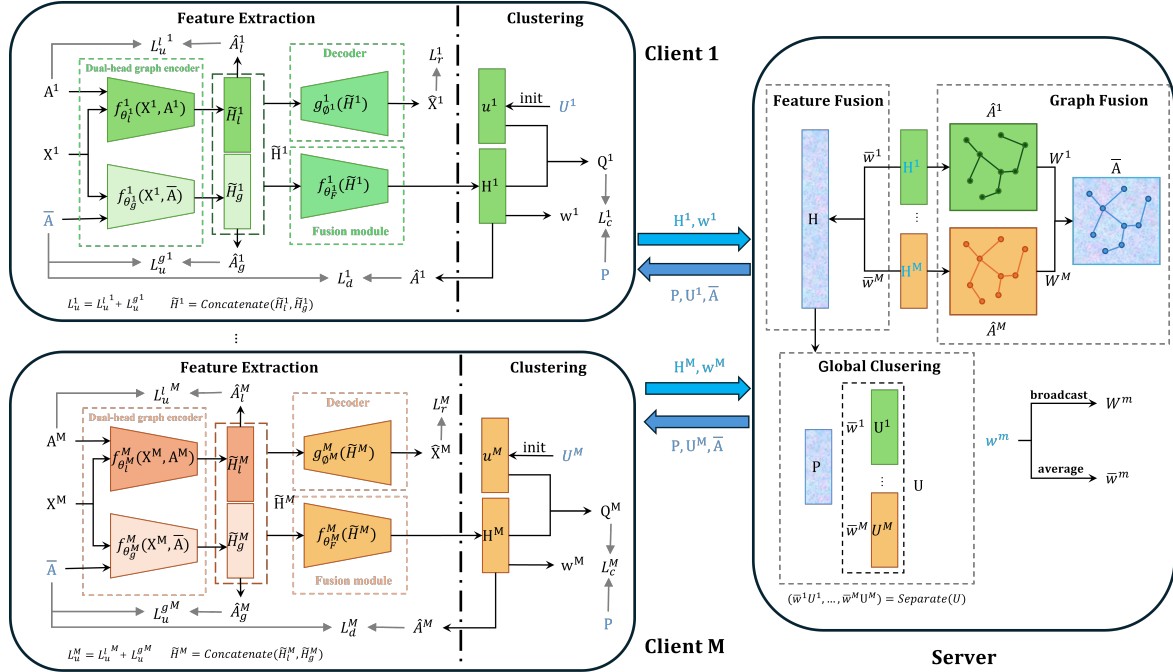

*Figure 1.* Overview of the FIMCFG framework. It contains $M$ clients and a server. (1) Client: $M$-th client consists of two processes: feature extraction and clustering. With the guidance of fused graph $\overline{A}$, client extracts underlying features $\widetilde{H}^M$ by dual-head graph encoder and high-level features $H^M$ by fusion module. Under the supervision of pseudo-labels $P$, the clustering centers $u^M$ are optimized by reducing the KL loss between soft distribution $Q^M$ and $P$. Based on clustered $H^M$, the silhouette coefficients $w^M$ are calculated as aggregation weights. (2) Server: the server performs feature fusion with $\{H^m\}_{m=1}^M$ to obtain global features $H$ and graph fusion with latent graph $\{\hat{A}^m\}_{m=1}^M$ of $\{H^m\}_{m=1}^M$ to obtain fused graph $\overline{A}$. Then the server perform K-means algorithm on $H$ to get global clustering centers $U$ and pseudo labels $P$.

missing samples are represented by zero vectors, the encoder automatically ignores them during computation. However, when using similarity-based calculations for the local adjacency matrix, missing samples represented as zero vectors are isolated from complete samples, making their feature computation infeasible. To address this issue, we propose a global graph structure migration technique, which replaces rows corresponding to missing samples in the local adjacency matrix with corresponding rows from the global adjacency matrix. This approach enriches the local adjacency matrix with global structural information, enabling the missing samples to estimate features based on adjacent complete samples.

### 3.2.2. TWO-LEVEL FEATURE EXTRACTION WITH FUSED GRAPH GUIDANCE

As demonstrated in Figure 1, the client's feature extraction consists of three parts: the dual-head graph encoder, the decoder and the fusion module. Guided by the globally fused graph, they are jointly trained to extract two-level features.

**Dual-Head Graph Encoder.** Dual-head Graph Encoder contains two parallel graph encoders. The samples $X^m$,

local graph $A^m$ and fused graph $\overline{A}$ are input to the dual-head graph encoder to extract two different underlying features $\widetilde{H}_l^m$ and $\widetilde{H}_g^m$. Specifically, stacked GCN layers are used as encoders and the global graph encoder $f_{\theta_g^m}^m$ and local graph encoder $f_{\theta_l^m}^m$ are used as two heads. Their mapping functions are : $f_{\theta_g^m}^m(X^m, \overline{A}) \longmapsto \widetilde{H}_g^m \in \mathbb{R}^{N \times \widetilde{d}_m}$ and $f_{\theta_l^m}^m(X^m, A^m) \longmapsto \widetilde{H}_l^m \in \mathbb{R}^{N \times \widetilde{d}_m}$, where $\widetilde{H}_g^m$ and $\widetilde{H}_l^m$ are the underlying features containing global and local view information, respectively. The computation process of the $t$-th layer of the local graph encoder is denoted as:

$$
\begin{aligned}
\widetilde{H}_{l\;(t)}^m = & \sigma\big(\widetilde{D}^{m-\frac{1}{2}} \widetilde{A}^m \widetilde{D}^{m-\frac{1}{2}} \widetilde{H}_{l\;(t-1)}^m W_{(t)}^m + b_{(t)}^m\big) \\
& + \widetilde{H}_{l\;(t-1)}^m,
\end{aligned} \tag{2}
$$

where $\sigma(\cdot)$ is activation function. $\widetilde{A}^m = A^m + I$ and $\widetilde{D}_{ii}^m = \sum_j \widetilde{A}_{ij}^m$. $I$ is the unit matrix, and $W_{(t)}^m$ and $b_{(t)}^m$ are the trainable parameters of the $t$-th layer. $A^m$ is the fixed local graph after global graph structure migration. Skip connections are introduced to prevent model degradation. The global graph encoder is computed in the same way with local graph encoder.

The dual-head graph encoder is identified by optimizing

the graph reconstruction loss such that features retain graph structure information. The latent graphs $\hat{A}_g^m \in [0,1]^{N \times N}$ and $\hat{A}_l^m \in [0,1]^{N \times N}$ are constructed by underlying features $\widetilde{H}_g^m$ and $\widetilde{H}_l^m$ according to Eq. (1). Their elements indicates the strength of the connection, 0 means no connection and 1 means strong connection. The graph reconstruction loss of the underlying features is calculated as follows:

$$L_u^m = \underbrace{\|\hat{A}_g^m - \overline{A}\|_2^2}_{L_u^{g\,m}} + \underbrace{\|\hat{A}_l^m - A^m\|_2^2}_{L_u^{l\,m}}. \tag{3}$$

**Decoder.** In order to enhance the robustness of the model, we set up a decoder with fully connected layer for the underlying features. The mapping function is $g_\phi^m : \widetilde{H}^m \in \mathbb{R}^{N \times 2\widetilde{d}_m} \longmapsto \hat{X}^m \in \mathbb{R}^{N \times D_m}$, where $\widetilde{H}^m$ represents the underlying feature matrix concatenated by $\widetilde{H}_g^m$ and $\widetilde{H}_l^m$, and $\hat{X}^m$ denotes the reconstructed samples. The content reconstruction loss is used to optimize:

$$L_r^m = \|X^m - \hat{X}^m\|_2^2. \tag{4}$$

Due to the rich view information, underlying features are suitable for reconstructing samples. Note that we compute the content reconstruction loss only for complete samples and use the predicted reconstructed samples to replace the missing samples at the end of training round.

**Fusion Module.** Since the two underlying features may contain conflicting components and redundant information which are not suitable for clustering, we introduce a fusion module that fuses the two underlying features and maps them into a low-dimensional high-level feature space. The fusion module uses fully connected layers with a mapping function: $f_{\theta_F^m}^m : \widetilde{H}^m \in \mathbb{R}^{N \times 2\widetilde{d}_m} \longmapsto H^m \in \mathbb{R}^{N \times d_m}$, where $H^m$ is high-level feature. In order to learn a consistent representation, globally fused graph is used to guide the training of fusion module. The consistent graph reconstruction loss for high-level features is computed as follows:

$$L_d^m = \|\hat{A}^m - \overline{A}\|_2^2, \tag{5}$$

where $\hat{A}^m \in [0,1]^{N \times N}$ is the latent graph constructed from high-level features $H^m$ by Eq. (1). The fusion module on each client aligns the fused high-level feature graph closer to the global graph, while the global graph is collaboratively updated by all clients. Through iterative updates, the model converges to a stable state, ultimately resolving conflicting components. Up to this point, we merge the graph reconstruction loss of dual-head graph encoder and fusion module as:

$$L_g^m = L_u^m + L_d^m. \tag{6}$$

During the communication phase, the client uploads high-level features to the server, which protects data privacy.

### 3.2.3. PSEUDO-LABEL-GUIDED CLUSTERING TRAINING

To obtain the clustering assignment, we construct a clustering layer $c_{u^m}^m$ with trainable parameters $\{u_j^m \in \mathbb{R}^{d_m}\}_{j=1}^K$, where $K$ represents the number of clustering and $u_j^m$ represents the clustering center of the $j$-th cluster on client $m$. The soft distribution $Q^m \in \mathbb{R}^{N \times K}$ is computed as follows:

$$q_{ij}^m = \frac{(1 + \|h_i^m - u_j^m\|_2^2)^{-1}}{\sum_{j=1}^K (1 + \|h_i^m - u_j^m\|_2^2)^{-1}}, \tag{7}$$

where $q_{ij}^m$ denotes the probability that the $i$-th sample is assigned to the $j$-th cluster, and $h_i^m \in \mathbb{R}^{d_m}$ denotes the high-level feature of the $i$-th sample. At the beginning of each training round, the client initializes the clustering layer with the global clustering center $U^m$ to align the classes. We use pseudo-labels from the server to supervise the training of the clustering layer such that different clients can obtain a consistent clustering structure. The KL divergence loss between the soft distribution $Q^m$ and the pseudo-labels $P$ is used to optimize:

$$L_c^m = D_{KL}(P\|Q^m) = \sum_{i=1}^N \sum_{j=1}^K p_{ij} \log \frac{p_{ij}}{q_{ij}^m}. \tag{8}$$

In addition, when not yet communicating, the client has no available pseudo-labels since the server has not yet performed aggregation. Therefore, during the first round of training, the client supervises the training of the clustering layer using the target distribution $P^m \in \mathbb{R}^{N \times K}$ instead of $P$. $P^m$ is computed by the following equation:

$$p_{ij}^m = \frac{(q_{ij}^m / \sum_j q_{ij}^m)^2}{\sum_j (q_{ij}^m / \sum_j q_{ij}^m)^2}. \tag{9}$$

Therefore, the optimization objective of client $m$ consists of three parts:

$$L^m = L_c^m + \gamma_1 L_g^m + \gamma_2 L_r^m, \tag{10}$$

where $\gamma_1$ and $\gamma_2$ are the trade-off parameters to balance the clustering loss, graph reconstruction loss and content reconstruction loss.

### 3.2.4. AGGREGATION WEIGHT

Silhouette comprehensively evaluates clustering quality by considering both intra-cluster cohesion and inter-cluster separation (Rousseeuw, 1987). It ranges between $[-1, 1]$, where values close to 1 indicate that samples are effectively clustered and well-separated from other clusters, reflecting strong clustering performance. Conversely, values close to $-1$ suggest incorrect cluster assignments and poor clustering quality. Thus we use it as the weights for server

aggregation. At the end of client's training, the client computes the silhouettes $w^m \in \mathbb{R}^N$ of the high-level feature and sends them to the server. The silhouettes for the $i$-th sample are calculated as follows:

$$w_i^m = \frac{b(i) - a(i)}{\max\{a(i), b(i)\}}. \tag{11}$$

Assume that the $i$-th sample is assigned to cluster $\hat{k}$, then $a(i)$ denotes the average distance of sample $i$ from other samples belonging to the same cluster $\hat{k}$. We use $d(i, k)$ to denote the average distance of sample $i$ from all samples belonging to cluster $k$, and then $b(i) = \min\limits_{k \neq \hat{k}} d(i, k)$.

### 3.2.5. PRE-TRAINING

Since there is no fused graph available at the first round of training, the clients need to conduct pre-training to extract features for the server to perform graph fusion. For pre-training, the client only optimizes the local graph encoder $f_{\theta_l^m}^m$ to obtain the local underlying features $\widetilde{H}_l^m$, and uploads $\widetilde{H}_l^m$ to the server for graph fusion. The local graph reconstruction loss is used to optimize the local graph encoder $f_{\theta_l^m}^m$:

$$L_{pre}^m = \|\tau_k(\hat{A}_l^m) - A^m\|_2^2, \tag{12}$$

where $\hat{A}_l^m \in [0, 1]^{N \times N}$ indicates the latent graph constructed from $\widetilde{H}_l^m$ mentioned above. $A^m \in \{0, 1\}^{N \times N}$ is the incomplete local graph before global graph structure migration. $\tau_k(\cdot)$ is a mask function that retain only the largest $k$ elements in each row of the latent graph, and set the rest to zero. Only the strongest $k$ edges are adopted to solve the incompleteness problem.

### 3.3. Server Global Aggregation

The server receives the high-level features $\{H^m\}_{m=1}^M$ and weights $\{w^m\}_{m=1}^M$ from the clients and performs graph fusion, feature fusion and global clustering.

**Graph Fusion.** To fuse the graphs, the server extracts the latent graph $\{\hat{A}^m \in [0, 1]^{N \times N}\}_{m=1}^M$ from $\{H^m\}_{m=1}^M$ by Eq. (1). After that, the server fuses the latent graph according to $\{w^m\}_{m=1}^M$. Specifically, it fuses the latent graphs as follows:

$$\overline{A} = f_k\left(\frac{1}{M} \sum_{m=1}^M (W^m \odot \hat{A}^m)\right), \tag{13}$$

where $\overline{A} \in \{0, 1\}^{N \times N}$ denotes the fused graph, $\odot$ denotes Hadamard product. $f_k(\cdot)$ denotes the function that sets the largest $k$ elements of each row in the matrix to 1 and the rest to 0. $W^m \in \mathbb{R}^{N \times N}$ denotes the weight matrix by broadcasting $w^m \in \mathbb{R}^N$.

**Feature Fusion.** The feature aggregation is conducted as follows:

$$H = [\overline{w}^{1'} H^1, \overline{w}^{2'} H^2, \cdots, \overline{w}^{M'} H^M], \tag{14}$$

where $H \in \mathbb{R}^{N \times \sum_{m=1}^M d_m}$ denotes the global feature, and $[\cdot, \cdots, \cdot]$ stands for matrix concatenation operation. $\overline{w}^{m'} = 1 + \log(1 + \frac{\overline{w}^m}{\sum_{m=1}^M |\overline{w}^m|})$, where $\overline{w}^m$ is the view silhouette coefficient and is calculated by $\overline{w}^m = \frac{1}{N} \sum_{i=1}^N w_i^m$.

**Global Clustering.** K-means algorithm is used to cluster $H$ to obtain the global clustering center $U = [\overline{w}^{1'} U^1, \overline{w}^{2'} U^2, \cdots, \overline{w}^{M'} U^M]$, where $U^m$ denotes the global clustering center of the $m$-th view. One point worth noting is that $\{U^m\}_{m=1}^M$ is aligned in classes. Based on the Student-t distribution, the soft distribution for each sample is:

$$s_{ij} = \frac{(1 + \|h_i - U_j\|_2^2)^{-1}}{\sum_{j=1}^K (1 + \|h_i - U_j\|_2^2)^{-1}}, \tag{15}$$

where $h_i$ denotes the global features of the $i$-th sample and $U_j$ denotes the $j$-th global clustering center. By sharpening the soft distribution, the pseudo-label $P$ is obtained as follows:

$$p_{ij} = \frac{(\frac{s_{ij}}{\sum_j s_{ij}})^2}{\sum_j (\frac{s_{ij}}{\sum_j s_{ij}})^2}. \tag{16}$$

Finally, the clustering labels for each sample is obtained. The category $y_i$ for the $i$-th sample is:

$$y_i = \arg\max_k p_{ik}. \tag{17}$$

### 3.4. Algorithm optimization

Algorithm 1 details the optimization process of FIMCFG consisting of two parts: clients and server. The client trains the local model in parallel, which pre-trains to extract local features before communication and uploads them to the server for graph fusion. In subsequent rounds of training, the clients extract the two-level features under the guidance of $\overline{A}$ and perform clustering with the supervision of $P$. The server fuses the features and latent graphs from the clients and performs global clustering to obtain the global clustering results. The client and server iterate alternately for $T$ rounds.

## 4. Experiments

### 4.1. Experimental Settings

#### 4.1.1. DATASETS AND METRICS

Our experiments were conducted on four widely used multi-view datasets. Specifically, Scene-15 (Lazebnik et al., 2006; Fei-Fei & Perona, 2005) consists of 4485 scene images classified into 15 classes, with each sample represented by three

*Table 1.* **Experimental results on the four datasets. The best results in each column are shown in bold and the second best results are underlined.** $\delta = 0$ **indicates complete while** $\delta = 0.5$ **indicates incomplete.**

| $\delta$ | Methods | HW | | | Scene-15 | | | Landuse-21 | | | 100Leaves | | |
|---|---|---|---|---|---|---|---|---|---|---|---|---|---|
| | | ACC | NMI | ARI | ACC | NMI | ARI | ACC | NMI | ARI | ACC | NMI | ARI |
| 0 | FCUIF | 0.965 | 0.923 | 0.925 | 0.471 | 0.444 | 0.293 | 0.274 | 0.302 | 0.130 | 0.910 | 0.964 | 0.878 |
| | FedMVFPC | 0.407 | 0.528 | 0.329 | 0.304 | 0.327 | 0.173 | 0.201 | 0.205 | 0.074 | 0.187 | 0.574 | 0.137 |
| | HFMVC | 0.789 | 0.732 | 0.666 | 0.367 | 0.381 | 0.217 | 0.237 | 0.272 | 0.096 | 0.709 | 0.880 | 0.624 |
| | DIMVC | 0.446 | 0.533 | 0.381 | 0.350 | 0.309 | 0.179 | 0.243 | 0.301 | 0.109 | 0.825 | 0.922 | 0.753 |
| | DSIMVC | 0.759 | 0.756 | 0.661 | 0.281 | 0.299 | 0.146 | 0.177 | 0.173 | 0.049 | 0.401 | 0.725 | 0.294 |
| | GIGA | 0.807 | 0.853 | 0.756 | 0.221 | 0.263 | 0.041 | 0.131 | 0.257 | 0.017 | 0.742 | 0.877 | 0.483 |
| | GIMVC | 0.935 | 0.886 | 0.874 | 0.426 | 0.465 | 0.279 | 0.258 | 0.337 | 0.114 | 0.857 | 0.952 | 0.819 |
| | CDIMC-net | 0.861 | 0.890 | 0.827 | 0.347 | 0.421 | 0.198 | 0.184 | 0.236 | 0.054 | 0.799 | 0.938 | 0.751 |
| | MRL_CAL | 0.478 | 0.526 | 0.337 | 0.194 | 0.168 | 0.069 | 0.163 | 0.169 | 0.045 | 0.224 | 0.587 | 0.126 |
| | Ours | **0.976** | **0.946** | **0.948** | **0.475** | **0.472** | **0.309** | **0.304** | **0.350** | **0.153** | **0.960** | **0.982** | **0.942** |
| 0.5 | FCUIF | 0.917 | 0.840 | 0.827 | 0.410 | 0.378 | 0.234 | 0.232 | 0.244 | 0.098 | 0.604 | 0.764 | 0.431 |
| | FedMVFPC | 0.313 | 0.364 | 0.192 | 0.203 | 0.199 | 0.071 | 0.134 | 0.123 | 0.025 | 0.134 | 0.397 | 0.030 |
| | HFMVC | 0.516 | 0.433 | 0.272 | 0.221 | 0.216 | 0.096 | 0.166 | 0.181 | 0.047 | 0.325 | 0.622 | 0.177 |
| | DIMVC | 0.322 | 0.255 | 0.151 | 0.310 | 0.261 | 0.143 | 0.226 | 0.278 | 0.099 | 0.579 | 0.731 | 0.380 |
| | DSIMVC | 0.729 | 0.687 | 0.586 | 0.260 | 0.267 | 0.125 | 0.172 | 0.169 | 0.048 | 0.295 | 0.616 | 0.171 |
| | GIGA | 0.764 | 0.730 | 0.594 | 0.146 | 0.127 | 0.008 | 0.182 | 0.279 | 0.025 | 0.418 | 0.649 | 0.055 |
| | GIMVC | 0.911 | 0.838 | 0.825 | 0.385 | 0.373 | 0.218 | 0.228 | 0.273 | 0.085 | 0.688 | 0.842 | 0.555 |
| | CDIMC-net | 0.858 | 0.861 | 0.792 | 0.217 | 0.268 | 0.067 | 0.122 | 0.161 | 0.020 | 0.330 | 0.643 | 0.207 |
| | MRL_CAL | 0.358 | 0.370 | 0.191 | 0.189 | 0.150 | 0.065 | 0.162 | 0.168 | 0.044 | 0.145 | 0.434 | 0.052 |
| | Ours | **0.952** | **0.897** | **0.896** | **0.444** | **0.420** | **0.264** | **0.293** | **0.323** | **0.139** | **0.802** | **0.886** | **0.696** |

---

**Algorithm 1** Optimization algorithm for FIMCFG

**Require:** Data with $M$ views $\{X^m\}_{m=1}^M$ distributed across $M$ clients, number of clusters $K$, number of communication rounds $T$, number of training rounds $E$.

**Ensure:** Clustering results $Y = \{y_1, y_2, \cdots, y_N\}$.

1: Pretrain the clients in parallel via Eq. (12).
2: Upload $\widetilde{H}_l^m$ to the server.
3: Calculate the fused graph $\overline{A}$.
4: Distribute $\overline{A}$ to the clients.
5: **while** not reaching T **do**
6:   **for** $m = 1$ to $M$ in parallel **do**
7:     Migrate the global graph structure.
8:     **while** not reaching E **do**
9:       Update $H^m$ by optimizing Eq. (10).
10:     **end while**
11:     Use $\hat{X}^m$ to replace missing samples.
12:     Calculate silhouette coefficients $w^m$.
13:     Upload $H^m$ and $w^m$ to the server.
14:   **end for**
15:   Update $\overline{A}$ by Eq. (13).
16:   Update $H$ by Eq. (14).
17:   Obtain $\{U^m\}_{m=1}^M$ by K-means.
18:   Obtain $P$ by Eq. (16).
19:   Distribute $\overline{A}$, $\{U^m\}_{m=1}^M$, $P$ to clients.
20: **end while**
21: Calculate the clustering label by Eq. (17).

views. HandWritten (HW)[1] contains 2000 samples in ten numeric categories, each consisting of six views. Landuse-21 (Yang & Newsam, 2010) consists of 2100 satellite images in 21 categories, 100 images per category, represented by three views. 100leaves[2] consists of 1600 image samples of 100 plants, each represented by three different views. Details of the dataset are presented in Table 2.

In the federated learning settings, multiple views of these datasets are distributed across different clients, each containing one of the views and isolated from each other. We define the data missing rate as $\delta = 1 - \frac{N_c}{N}$, where $N_c$ represents the number of overlapping samples in all clients and $N$ represents the number of all samples in the dataset. To construct the incomplete dataset, we randomly selected $N - N_c$ samples in the dataset, and set the $n$ views of each sample among them to 0, where $n = [n']$, $[\cdot]$ denotes rounding operation and $n' \sim U(1, M-1)$.

We use three commonly-used metrics to evaluate the effectiveness of clustering, i.e., clustering accuracy (ACC), normalized mutual information (NMI), and adjusted random index (ARI). A higher value of each metric indicates a better clustering performance.

---

[1]https://archive.ics.uci.edu/dataset/72/multiple+features
[2]https://archive.ics.uci.edu/ml/datasets/One-hundred+plant+species+leaves+data+set

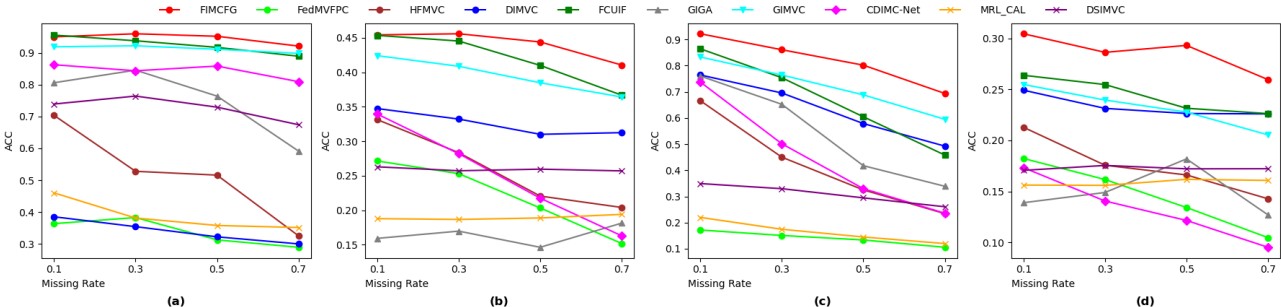

*Figure 2.* Accuracy on four datasets with different missing rates. (a) HW, (b) Scene-15, (c) 100Leaves, (d) Landuse-21.

*Table 2.* Description of the datasets.

| Datasets | Samples | Views | Distribution of dimensions | | | | | | Classes |
|----------|---------|-------|-----|-----|-----|-----|-----|-----|---------|
| Scene-15 | 4485 | 3 | 20 | 59 | 40 | - | - | - | 15 |
| HandWritten | 2000 | 6 | 240 | 76 | 216 | 47 | 64 | 6 | 10 |
| Landuse-21 | 2100 | 3 | 20 | 59 | 40 | - | - | - | 21 |
| 100leaves | 1600 | 3 | 64 | 64 | 64 | - | - | - | 100 |

### 4.1.2. COMPARED METHODS

In order to verify the effectiveness and superiority of our method, seven state-of-the-art methods including three federated MVC methods and six centralized IMVC methods are chosen to be compared methods. They are listed as follows:

- FCUIF(Ren et al., 2024) utilizes sample commonality, view versatility, and adaptive imputation techniques to address unaligned and incomplete data under federated setting.

- FedMVFPC (Hu et al., 2024) is a federated learning method designed for privacy-preserving multiview fuzzy clustering.

- HFMVC (Jiang et al., 2024) is a heterogeneity-aware federated deep multi-view clustering method that leverages contrastive learning to explore consistency and complementarity across multi-view data.

- DIMVC(Xu et al., 2022) is a imputation-free and fusion-free deep IMVC framework.

- DSIMVC(Tang & Liu, 2022) proposes a bi-level optimization framework that dynamically fills missing views using learned neighbor semantics.

- GIGA(Yang et al., 2024) adaptively estimates the factual weight of each available view to mitigate the effects of missing views.

- GIMVC(Bai et al., 2024) is a graph-guided, imputation-free incomplete multi-view clustering method.

- CDIMC-net(Wen et al., 2020) combines view-specific deep encoders and graph embedding strategy to capture the high-level features and local structure of each view.

- MRL_CAL(Wang et al., 2024a) utilizes joint learning of features in different subspaces for data recovery, consistent representation and clustering.

We compared FIMCFG with baselines at two missing rate settings: $\delta = 0$ (complete) and $\delta = 0.5$ (incomplete).

### 4.2. Experimental Results

Table 1 shows the clustering results of FIMCFG against the compared methods in both complete and incomplete scenarios. It can be observed that our method outperforms all the compared methods on all the datasets in both scenarios, demonstrating the superiority of our method. In particular, in the missing settings, our method significantly outperforms the second-ranked compared method on all the four datasets.

To further investigate the robustness of our method to the missing rate, we conducted experiments on four datasets with different missing rates ranging from 0.1 to 0.7 with an interval of 0.2. As shown in Figure 2, our method outperforms all the compared methods at almost all the missing rate settings across all the four datasets, and this advantage becomes more and more significant as the missing rate increases. The results show that our method is robust to different missing rates, and can use global information to estimate the data distribution even with high missing rates.

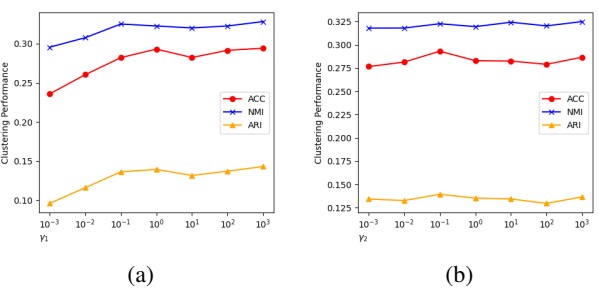

(a)                    (b)

*Figure 3.* Parameter sensitivity analysis on Landuse-21 when $\delta = 0.5$. (a) Clustering performance with different $\gamma_1$ values, (b) Clustering performance with different $\gamma_2$ values.

## 4.3. Model Analysis

### 4.3.1. ABLATION STUDY

To further investigate the effectiveness of the global information in fused graph, we conducted an ablation study to explore the impact of each component related to the fused graph on clustering results. The components related to the fused graph include: (A) global graph structure migration, (B) feature fusion module, (C) global graph guidance, and (D) global graph encoder. We remove each component to explore its effect. It should be noted that when we remove the global graph encoder, all these components will be not exist. The ablation study experimental results are shown in Table 3. It can be easily seen that each module plays an important role.

*Table 3.* Ablation study on Scene-15 when $\delta = 0.5$.

| Components | | | | Scene-15 | | |
|---|---|---|---|---|---|---|
| A | B | C | D | ACC | NMI | ARI |
| | ✓ | ✓ | ✓ | 0.431 | 0.415 | 0.258 |
| ✓ | | ✓ | ✓ | 0.357 | 0.407 | 0.205 |
| ✓ | ✓ | | ✓ | 0.412 | 0.414 | 0.251 |
| | | | | 0.247 | 0.210 | 0.095 |
| ✓ | ✓ | ✓ | ✓ | 0.444 | 0.420 | 0.264 |

### 4.3.2. PARAMETER ANALYSIS

During the training of clients, the total loss function defined by Eq. (10) has two hyperparameters $\gamma_1$ and $\gamma_2$ to trade-off the graph reconstruction loss and content reconstruction loss. We conducted the experiments with various settings of the two hyperparameters ranging from $10^{-3}$ to $10^3$ at $\delta = 0.5$, as shown in Figure 3. We observe that $\gamma_1$ in the range of $[10^{-2}, 10^3]$ is robust for clustering results. Too small $\gamma_1$ degrades the clustering performance due to the fact that it makes the encoder to ignore the global information of the fused graph. The clustering performance is not sensitivity to $\gamma_2$. Based on the experimental results, we recommend setting $\gamma_1$ to 1 and $\gamma_2$ to 0.1 for optimal performance.

### 4.3.3. HETEROGENEITY ANALYSIS

In order to study the effect of the data heterogeneity in clients on FIMCFG, we introduced the Dirichlet distribution in the construction of the incomplete dataset. The smaller the parameter $\alpha$ of Dirichlet, the larger the imbalance in the size of samples on the clients. We set up three scenarios with different data distributions: $\alpha = 10^{-2}$ (High), $\alpha = 1.0$ (Moderate) and random distribution (None) to illustrate three heterogeneous scenarios. The results are shown in Figure 4, which shows that our model is less affected by data heterogeneity and performs well even in highly heterogeneous scenarios.

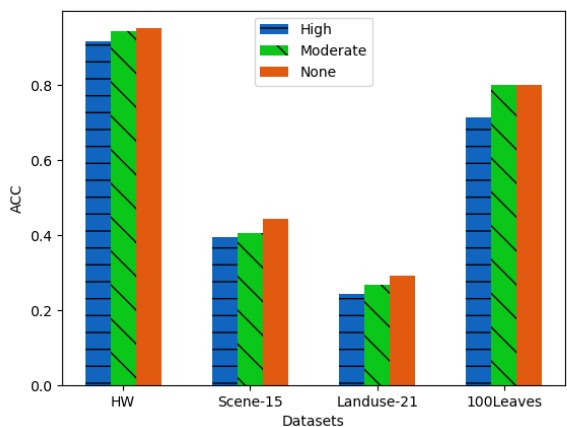

*Figure 4.* Sensitivity to imbalanced sample sizes in clients on four datasets with a missing rate of 0.5.

## 5. Conclution

In this paper, we propose a novel federated incomplete multi-view clustering method FIMCFG. We designed the global graph structure migration to correct the local graphs and estimates the missing features with GCN to tackle the missing data problem. Moreover, we designed the dual-head graph encoder and fusion module to extract high-level features by using the fused global graph. Experimental results verified the effectiveness and superiority of FIMCFG.

## Acknowledgments

This work is supported in part by the National Natural Science Foundation of China (No. 62276079), Young Teacher Development Fund of Harbin Institute of Technology IDGA10002071, Research and Innovation Foundation of Harbin Institute of Technology IDGAZMZ00210325, Key Research and Development Plan of Shandong Province 2021SFGC0104 and the Special Funding Program of Shandong Taishan Scholars Project.

## Impact Statement

This paper presents work whose goal is to advance the field of Machine Learning. There are many potential societal consequences of our work, none of which we feel must be specifically highlighted here.

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

# A. More Experiments of FIMCFG

**A.1.** We show more ablation study results on HW, Landuse-21 and 100Leaves in Table 4.

*Table 4.* Ablation study results on HW, Landuse-21 and Scene-15 with $\delta = 0.5$.

| Components | | | | HW | | | Landuse-21 | | | 100Leaves | | |
| --- | --- | --- | --- | --- | --- | --- | --- | --- | --- | --- | --- | --- |
| A | B | C | D | ACC | NMI | ARI | ACC | NMI | ARI | ACC | NMI | ARI |
| | ✓ | ✓ | ✓ | 0.950 | 0.896 | 0.892 | 0.252 | 0.277 | 0.114 | 0.758 | 0.870 | 0.650 |
| ✓ | | ✓ | ✓ | 0.930 | 0.889 | 0.876 | 0.213 | 0.284 | 0.084 | 0.792 | 0.885 | 0.687 |
| ✓ | ✓ | | ✓ | 0.912 | 0.865 | 0.847 | 0.282 | 0.325 | 0.138 | 0.782 | 0.880 | 0.679 |
| | | | | 0.556 | 0.543 | 0.366 | 0.175 | 0.179 | 0.051 | 0.430 | 0.666 | 0.231 |
| ✓ | ✓ | ✓ | ✓ | 0.952 | 0.897 | 0.896 | 0.293 | 0.323 | 0.139 | 0.802 | 0.886 | 0.696 |

**A.2.** We show the experimental results in NMI and ARI for all the methods under different missing rates ranging from 0.1 to 0.7 with an interval of 0.2. The results are shown in Figure 5 and Figure 6, respectively.

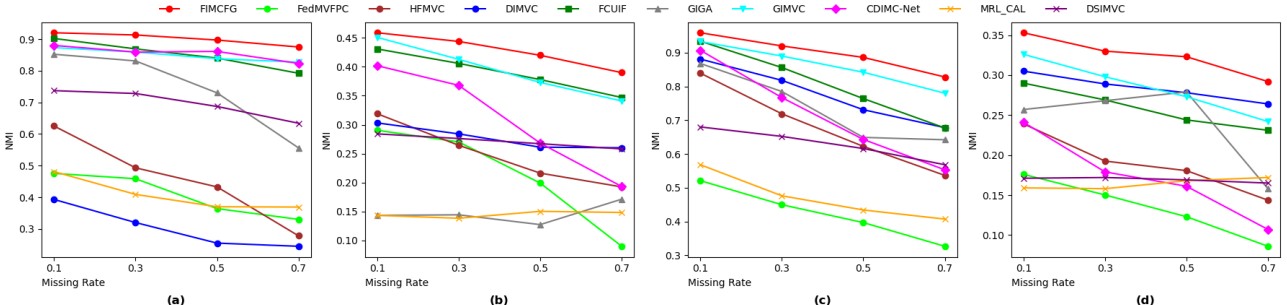

*Figure 5.* NMI on four datasets with different missing rates. (a) HW, (b) Scene-15, (c) 100Leaves, (d) Landuse-21.

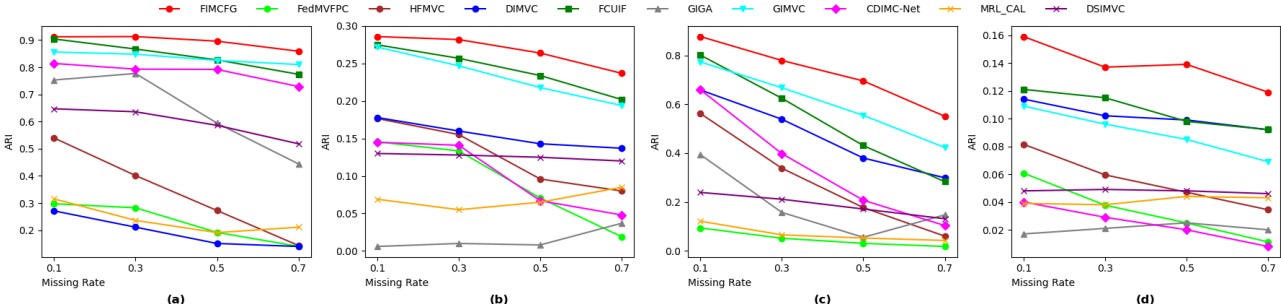

*Figure 6.* ARI on four datasets with different missing rates. (a) HW, (b) Scene-15, (c) 100Leaves, (d) Landuse-21.

**A.3.** We show more parameter analysis experiments on HW, Scene-15 and 100Leaves data sets. Parameter sensitivity analysis of $\gamma_1$ are shown in Figure 7. Parameter sensitivity analysis of $\gamma_2$ are shown in Figure 8.

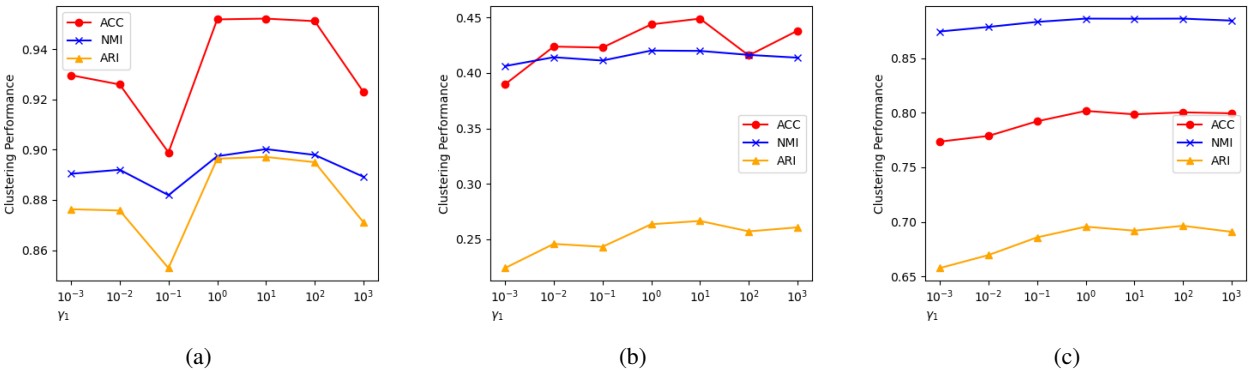

*Figure 7.* Parameter sensitivity analysis of $\gamma_1$ when $\delta = 0.5$ on three data sets: (a) HW, (b) Scene-15, (c) 100Leaves.

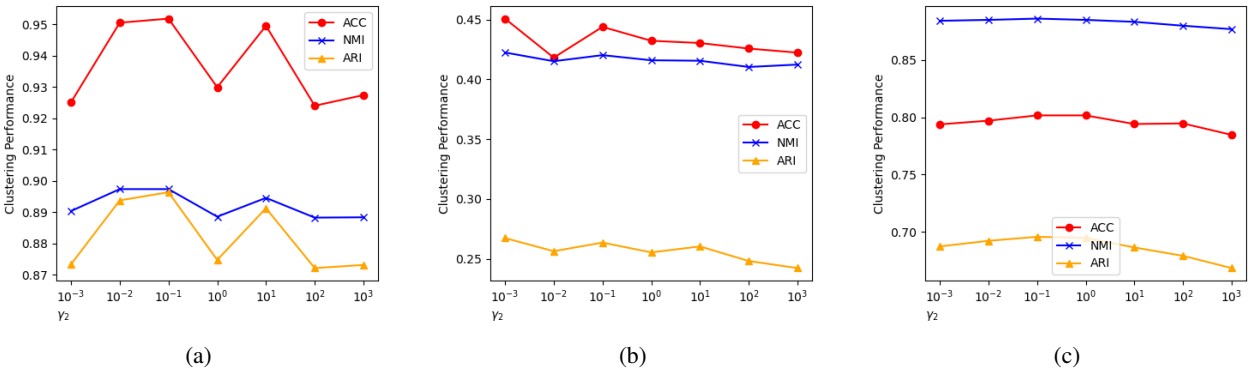

*Figure 8.* Parameter sensitivity analysis of $\gamma_2$ when $\delta = 0.5$ on three data sets: (a) HW, (b) Scene-15, (c) 100Leaves.

