# OpenReview forum: "Federated Incomplete Multi-view Clustering with Globally Fused Graph Guidance"
_ICML.cc/2025/Conference — ICML 2025 poster_

### Official Review · Reviewer_RBHX · 2025-03-05

**Overall Recommendation:** 4

**Summary:**

This paper presents a novel Federated Incomplete Multi-view Clustering method with globally Fused Graph guidance (FIMCFG), addressing the challenges of privacy preservation and data incompleteness in federated multi-view clustering framework. The main contribution of this work lies in its novel approach to handling incomplete multi-view data in a distributed setting while leveraging global information to improve the clustering performance.

**Claims And Evidence:**

Yes

**Essential References Not Discussed:**

[1] also explores the federated multi-view clustering method over distributed data. Their commons and differences may be worth to be introduced and discussed.

[1] Liu et al. Active-Passive Federated Learning for Vertically Partitioned Multi-view Data. Arxiv 2024.

**Experimental Designs Or Analyses:**

Yes. I have checked the comparison to existing methods, ablation study, parameter analysis and heterogeneity analysis. They well support the claims.

**Methods And Evaluation Criteria:**

Yes. The metrics, including ACC, NMI and ARI, are commonly used in clustering analysis.

**Other Comments Or Suggestions:**

Please see Section **Other Strengths And Weaknesses**

**Other Strengths And Weaknesses:**

Strength:

1. The design of dual-Head graph convolutional encoder is novel and interesting.

2. To address the missing data issue, the authors introduce a global graph structure migration technique. This method repairs the incomplete local graphs by leveraging the global graph structure, enabling the estimation of latent features to recover the missing data.

3. The paper is well organized and easy to follow.

Weaknesses:

1. Related works on incomplete multi-view clustering lacks: Incomplete multi-view clustering works should be introduced, since it is quite related to the proposed method.

2. The writing expression needs improvement: Although the paper organization is good, the paper writing still needs improvements. There are some grammar errors or typos, such as “Most” in “In summary, Most  of these approaches” in the third paragraph of Section Introduction should be “most”, “client extract underlying features” in caption of Figure 1 should be “client extracts underlying features”, please check throughout the whole paper and correct them.

3. Interaction between clients and server is not clear: What information are exchanged in Subsection “3.2. Client Local Training with Global Guidance” should be made more clear.

**Questions For Authors:**

Please see Section **Other Strengths And Weaknesses**

**Relation To Broader Scientific Literature:**

This paper propose a novel approach to group the incomplete multi-view data in a distributed setting. This is important but lacks exploration in depth in literature. So there would be a broad impact.

**Theoretical Claims:**

N/A. There is no theoretical claim.

---

> ### Author Rebuttal · Authors · 2025-03-31
>
> R1: Thanks for your good advice. Incomplete multi-view clustering are introduced in GNN based multi-view clustering and federated multi-view clustering. To make it more clear, we can extract these and introduce them together.
> R2: Thanks for your careful check. We'll correct all these errors or typos in the revised version if it is accepted finally.
> R3: Thanks for your helpful suggestion. During communication, the server transmits the fused graph  to the client to guide the upstream feature extraction process, the pseudo labels P to guide the training of the downstream clustering layer, and the cluster centers  to align the clustering layers of different clients. Meanwhile, the client uploads high-level features  and their corresponding weights  to the server for aggregation. We will elaborate further  on this in Subsection 3.2.

---

> > ### Comment · Reviewer_RBHX · 2025-04-03
> >
> > Thanks for your responses. They have addressed my concerns. So I recommend to accept.

---

> > > ### Author Response · Authors · 2025-04-09
> > >
> > > Thanks a lot for your acknowledgement. For weakness 1, we have supplemented the detailed reply. Since this weakness is similar with weakness 3 from Reviewer KxVe, we reply with the same response, please see the supplemented reply.

---

### Official Review · Reviewer_fzVj · 2025-03-07

**Overall Recommendation:** 3

**Summary:**

This paper provides a federated incomplete multi-view clustering approach to solve incomplete data and data privacy problem. Dual-head graph convolutional encoder is designed to extract the underlying features, the global graph structure migration is designed to repair incomplete local graphs to estimate the missing feature further. The extensive experimental comparison and analysis demonstrated the proposed approach works well and performs better than compared state-of-the-art methods.

**Claims And Evidence:**

The authors provide experiments and code to support the proposed method.

**Essential References Not Discussed:**

There are a few field-related papers that are not cited:
[1] An efficient federated multi-view fuzzy c-means clustering method.
[2] Efficient federated multi-view learning.
[3] Heterogeneity-Aware Federated Deep Multi-View Clustering towards Diverse Feature Representations.

**Experimental Designs Or Analyses:**

The experiments are varied and include performance analysis and model analysis. The selected datasets used are common multi-view datasets, but I would like to see more experiments on large-scale datasets and more federated multi-view approaches for comparison.

**Methods And Evaluation Criteria:**

Yes.

**Other Comments Or Suggestions:**

See Weaknesses.

**Other Strengths And Weaknesses:**

Strengths:
(1) By introducing the globally fused graph to guide the upstream feature extraction process, the model could exploit the global information when extracting features in distributed environment, which improves the clustering performance.
(2) With the global graph structure migration, the model is robust to different missing rates.
(3) The experiment in the work is sufficient to prove the validity of the model.
(4) The technical details of the method are detailed.

Weaknesses:
(1) How the fusion module deals with the conflicting components should be illustrated in detail. This can help the readers to understand this point well.
(2) How to recover the missing feature values needs more illustration. Is it implemented by optimizing the content reconstruction loss?
(3) The writing of this paper can be improved. For example, Eq (2) is out of the content area. The singular and plural forms of some words are wrong.
(4) More experiments on large-scale datasets and more federated multi-view approaches for comparison are desired.

**Questions For Authors:**

See Weaknesses.

**Relation To Broader Scientific Literature:**

This paper is useful for further exploration of federated multi-view clustering.

**Theoretical Claims:**

I checked the formulas in the methods section and they are all correct.

---

> ### Author Rebuttal · Authors · 2025-03-31
>
> R1: Thanks for your helpful advice. The fusion module on all clients brings the fused high-level feature’s graph structure closer to the global graph. Conversely, the global graph is updated collaboratively by all clients. Through such iterations, the model converges to a stable state, eventually eliminating conflicting parts.
> R2: The missing features are automatically estimated utilizing the GCN encoder and the global graph structure migration, as shown in the response R1 to Reviewer 1. Our content reconstruction loss is primarily designed to enhance the robustness of the model and to learn the distribution of missing samples.
> R3: Thank you for your useful suggestions. We will rectify the errors  in the revised version and conduct a thorough check.
> R4: Thanks for your good advice. We'll add some more experiments or some deep discussion in the revised version.

---

> > ### Comment · Reviewer_fzVj · 2025-04-03
> >
> > I read the authors' responses to the questions posed by me and other reviewers, and I wish that the authors, when responding to the reviewers' comments about additions (e.g., reviewer KxVe's R2, reviewer UkiW's R6, reviewer fzVj's R4, and reviewer RBHX's R1), would have shown some preliminary results and explanations instead of just a sentence that the authors would revise it. Whether or not these suggestions can actually be implemented is probably what the reviewers are eager to know at the moment, and will influence our judgment of the paper.

---

> > > ### Author Response · Authors · 2025-04-09
> > >
> > > Thanks for your helpful suggestions. For your weakness 4, we added two federated multi-view approaches HFMVC[1] and FedMVFPC[2] to conduct experiments, and the results are shown as below:
> > >
> > > HFMVC
> > >
> > > \delta=0, complete case:
> > >
> > > Data set,              ACC,    NMI,    ARI
> > >
> > > Scene-15,          0.367,  0.381,  0.216
> > >
> > > HandWritten,     0.788,  0.732,  0.666
> > >
> > > 100Leaves,       0.708,  0.880,  0.623
> > >
> > > Landuse-21,     0.236,  0.271,  0.095
> > >
> > > ............................................................
> > >
> > > \delta=0.5, incomplete case:
> > >
> > > Data set,              ACC,    NMI,    ARI
> > >
> > > Scene-15,          0.220,  0.216,  0.095
> > >
> > > HandWritten,     0.516,  0.432,  0.272
> > >
> > > 100Leaves,       0.325,  0.622,  0.176
> > >
> > > Landuse-21,     0.166,  0.180,  0.047
> > >
> > > -------------------------------------------------
> > >
> > > FedMVFPC
> > >
> > > \delta=0, complete case:
> > >
> > > Data set,              ACC,    NMI,    ARI
> > >
> > > Scene-15,          0.304,  0.326,  0.172
> > >
> > > HandWritten,     0.406,  0.527,  0.328
> > >
> > > 100Leaves,       0.186,  0.574,  0.136
> > >
> > > Landuse-21,     0.200,  0.204,  0.073
> > >
> > > ............................................................
> > >
> > > \delta=0.5, incomplete case:
> > >
> > > Data set,              ACC,    NMI,    ARI
> > >
> > > Scene-15,          0.203,  0.198,  0.007
> > >
> > > HandWritten,     0.312,  0.364,  0.191
> > >
> > > 100Leaves,       0.133,  0.396,  0.030
> > >
> > > Landuse-21,     0.134,  0.122,  0.024
> > >
> > > From these results and compared with Table 1 in the paper, HFMVC outperformed FedMVFPC, but both of them perform worse than our proposed method. In addition, we also conducted the experiments with \delta = 0.1 0.5 0.7, due to the page limit, we didn't show here.
> > > For other reviewers' comments you mentioned, they suggest adding a data statistics table, adding related works on incomplete multi-view  and incomplete multi-view clustering, there are no problem to tackle them. We have added the detailed reply.
> > >
> > > [1]Jiang X, Ma Z, Fu Y, et al. Heterogeneity-Aware Federated Deep Multi-View Clustering towards Diverse Feature Representations[C]//Proceedings of the 32nd ACM International Conference on Multimedia. 2024: 9184-9193.
> > > [2]Hu X, Qin J, Shen Y, et al. An efficient federated multiview fuzzy c-means clustering method[J]. IEEE Transactions on Fuzzy Systems, 2023, 32(4): 1886-1899.

---

### Official Review · Reviewer_UkiW · 2025-03-13

**Overall Recommendation:** 4

**Summary:**

The authors proposed a federated incomplete multi-view clustering framework named FIMCFG. It designed a dual-head graph convolutional encoder at the client to extract the global and view-specific information. With the guidance of the fused graph, high-level features are used to conduct clustering under the supervision of pseudo-label. As the federated learning framework, it preserves the privacy well. Incomplete data problem is addressed with the fused graph and graph convolutional operation. The idea of this work is novel. It offers a tool to deal with incomplete multi-view clustering in federated learning framework. The experiments show the effectiveness and superiority of the proposed method.

**Claims And Evidence:**

Yes

**Essential References Not Discussed:**

None.

**Experimental Designs Or Analyses:**

Yes

**Methods And Evaluation Criteria:**

Yes, the authors adopted several evaluation metrics (ACC, NMI, ARI) for performance evaluation.

**Other Comments Or Suggestions:**

Despite key works are discussed, it is recommended to conduct a wider range of literature reviews.

**Other Strengths And Weaknesses:**

Strength:
1. Besides clustering phase, the global information are also mined in feature extracting with the designed dual-head GCN encoder. This is interesting and novel.
2. The global graph structure migration is proposed to fill in incomplete local graph, which is further used to estimate the missing values in original data.
3. This work considers the missing data and data privacy problems together.
4. The experiments are rich. Besides compared experiments, ablation study, parameter analysis, the authors conducted the experiments with data under different missing rate to show its performance. In addition, The heterogeneity analysis shows the methods’ performance on different data heterogeneity scenarios, this is important in federated learning.
Weakness:
1. Silhouette should be explained in detail.
2. Some related works on incomplete multi-view should be added and discussed.
3. The expression needs further polishing.

**Questions For Authors:**

Refer to the weakness.

**Relation To Broader Scientific Literature:**

The authors explored the federated incomplete multi-view clustering by developing a better way for global information fusion.

**Theoretical Claims:**

The authors mainly evaluated the work via a number of experiments.

---

> ### Author Rebuttal · Authors · 2025-03-31
>
> R1-4: Thanks for your acknowledgement.
> R5: Silhouette comprehensively considers the similarity between samples within a cluster and the distance between different clusters. It evaluates the  clustering quality based on two factors: cohesion and separation. Silhouette ranges between [-1, 1], where a value close to 1 indicates that the samples are well clustered within their respective cluster and are well separated from other clusters, demonstrating good clustering performance. Conversely, a value close to -1 suggests that the samples may have been incorrectly assigned to clusters, resulting in poor clustering performance. We will include the aforementioned explanation of the silhouette in the revised Section 3.2.4 and provide a detailed exaplanation of within-cluster cohesion, between-cluster separation, and the formula for silhouette.
> R6: Thanks for your useful suggestion. We'll add some recent related works on incomplete multi-view learning and discuss them in the final version if it is accepted.
> R7: Thanks for your advice. We'll check throughout the whole paper and improve the language.

---

### Official Review · Reviewer_KxVe · 2025-03-14

**Overall Recommendation:** 4

**Summary:**

The work proposes a novel GCN-based federated incomplete multi-view clustering framework. The information propagation limitation problem is solved by introducing the globally fused graph guidance when extracting features. The global graph structure migration is proposed in this paper. The incomplete data problem is solved by repairing the incomplete local graph with the fused graph. An adaptive weighted aggregation approach is developed, which could automatically adjust the importance of each view when conducting feature fusion and graph fusion. Experimental results demonstrate the effectiveness of the proposed method.

**Claims And Evidence:**

The claims of the paper are well supported by the experimental results.

**Essential References Not Discussed:**

The references in this paper are relatively sufficient, but it may be beneficial to consider citing some recent studies on federated multi-view learning.

**Experimental Designs Or Analyses:**

The experiments are conducted on four widely used multi-view datasets, providing comprehensive and sufficient results.

**Methods And Evaluation Criteria:**

The proposed federated incomplete multi-view clustering method effectively tackles the problems of global information exploration and missing data in federated multi-view clustering tasks.

**Other Comments Or Suggestions:**

Please refer to the aforementioned strengths and weaknesses.

**Other Strengths And Weaknesses:**

Strengths:
1. The idea of dual-head graph convolutional encoder to extract the features and globally fused graph guidance is novel and interesting.
2. The proposed method deals with the missing data well, demonstrating strong performance even with high missing data rates, making it suitable for real-world applications where data incompleteness is common.
3. By operating in a federated learning framework, the method ensures that raw data remains on local devices, addressing the data privacy concerns.
4. The authors conduct extensive experiments on multiple datasets, demonstrating the superiority of their method over existing approaches.

Weaknesses:
1. How the  global graph structure migration and encoders solve the incomplete problem should be explained in detail.
2. It is suggested to introduce the data statistics in the experiments in a table.
3. More federated multi-view clustering works are suggested to introduced to make the related works rich.

**Questions For Authors:**

Please refer to the aforementioned weaknesses.

**Relation To Broader Scientific Literature:**

This paper is closely related to existing studies on federated multi-view clustering and builds upon them by proposing a novel globally fused graph guidance method.

**Theoretical Claims:**

The proposed method, which includes Client Local Training with Global Guidance and Server Global Aggregation, is correctly designed and implemented.

---

> ### Author Rebuttal · Authors · 2025-03-31
>
> R1：Under the effect of the GCN encoder on clients, each sample estimates its single-view features using its own attribute values and those of its neighboring nodes. Since we fill the missing samples with zeros vector, the encoder automatically ignores these missing samples during computation. However, for the missing samples, we use similarity-based calculations for the local adjacency matrix. As a result, the missing samples (represented as zero vectors) are not adjacent to any complete samples, making it impossible to compute their features. To tackle this issue, we propose global graph structure migration, where the rows corresponding to missing samples in the local adjacency matrix are replaced with rows of the global adjacency matrix that integrates multi-view information. This complements the local adjacency matrix with the graph structure of missing samples, enabling them to estimate their features using adjacent complete samples.
> R2：Thank you for your suggestion. We will add the dataset size, the number of views, and the dimensions of each view in tabular form in the subsection 4.1.1 of the revised version.
> R3：Thanks for your advice. We'll add the following works to enrich the federated multi-view clustering related works.
> [1] Hu X, Qin J, Shen Y, et al. An efficient federated multiview fuzzy c-means clustering method[J]. IEEE Transactions on Fuzzy Systems, 2023, 32(4): 1886-1899.
> [2] Huang S, Shi W, Xu Z, et al. Efficient federated multi-view learning[J]. Pattern Recognition, 2022, 131: 108817.
> [3] Chen X, Ren Y, Xu J, et al. Bridging Gaps: Federated Multi-View Clustering in Heterogeneous Hybrid Views[J]. Advances in Neural Information Processing Systems, 2024, 37: 37020-37049.

---

> > ### Comment · Reviewer_KxVe · 2025-04-06
> >
> > The author's response effectively resolved my concerns, and I agree to accept this paper.

---

> > > ### Author Response · Authors · 2025-04-09
> > >
> > > Thanks for your acknowledgement. For weakness 2, we supplement the data statics table as below:
> > >
> > > Datasets, 	#Samples, 	#Views, 	#dimensions of each view, 	   #Classes
> > >
> > > Scene-15, 	4485,		3,		[20,	59,	40],					15
> > >
> > > HandWritten,	2000,		6,		[240,	76,	216,	47,	64,	6],		10
> > >
> > > LandUse-21,	2100,		3,		[20,	59,	40],					21
> > >
> > > 100leaves,	1600,		3,		[64,	64,	64],					100
> > >
> > > For weakness 3, we supplement the following related works descriptions.
> > >
> > > In real-world applications, multi-view data often suffer from missing data problem due to some uncontrollable factors in data collection, transmission, or storage. Incomplete Multi-View Clustering (IMVC) addresses this challenge by learning the robust clustering structures from partially observed multi-view data. Recent advances in deep learning have significantly enhanced IMVC performance, with various innovative approaches proposed. For instance, Completer [1] leverages the autoencoders to maximize  the cross-view mutual information via contrastive learning, ensuring view-consistent representations. Additionally, it employs the dual prediction to minimize the conditional entropy, effectively recovering the missing views. Another approach, based on variational autoencoders [2], utilizes the Product-of-Experts (PoE) method to aggregate multi-view information, deriving a shared latent representation to handle incompleteness. Further improving data recovery, AIMC [3] integrates the element-wise reconstruction with Generative Adversarial Networks (GANs) to generate the plausible missing data. More recently, Graph Neural Networks (GNNs) have been introduced to IMVC, capitalizing on their ability to model relational data. For example, ICMVC [4] tackles the missing data through multi-view consistency relation transfer combined with Graph Convolutional Networks (GCNs). Similarly, CRTC [5] introduces a cross-view relation transfer completion module, where GNNs infer missing data based on transferred relational graphs.
> > >
> > > [1]Lin Y, Gou Y, Liu Z, et al. Completer: Incomplete multi-view clustering via contrastive prediction[C]//Proceedings of the IEEE/CVF conference on computer vision and pattern recognition. 2021: 11174-11183.
> > > [2]Xu G, Wen J, Liu C, et al. Deep variational incomplete multi-view clustering: Exploring shared clustering structures[C]//Proceedings of the AAAI conference on artificial intelligence. 2024, 38(14): 16147-16155.
> > > [3] Xu C, Guan Z, Zhao W, et al. Adversarial incomplete multi-view clustering[C]//IJCAI. 2019, 7: 3933-3939.
> > > [4] Chao G, Jiang Y, Chu D. Incomplete contrastive multi-view clustering with high-confidence guiding[C]//Proceedings of the AAAI conference on artificial intelligence. 2024, 38(10): 11221-11229.
> > > [5] Wang Y, Chang D, Fu Z, et al. Incomplete multiview clustering via cross-view relation transfer[J]. IEEE Transactions on Circuits and Systems for Video Technology, 2022, 33(1): 367-378.

---

### Decision · Program_Chairs · 2025-05-01

**Decision:**

Accept (poster)

**Comment:**

This paper addresses an important challenge in federated multi-view clustering, namely the dual issues of incomplete data and privacy preservation in distributed settings. The authors propose a method that leverages a dual-head graph convolutional encoder to extract both global and view-specific features. A key novelty is the use of a globally fused graph to guide feature extraction and repair incomplete local graphs, thereby facilitating effective clustering under both complete and missing data conditions. Experimental evaluations on several widely used multi-view datasets, across varying missing rates, reinforce the proposed method’s superiority over a range of state-of-the-art techniques.

All reviewers have converged toward a positive evaluation, with overall recommendations ranging from “Accept” (score of 4) to “Weak Accept” (score of 3). Based on the strength of the contributions, the novelty in addressing both privacy preservation, data incompleteness, and robust experimental validations, the meta-review recommends acceptance of this paper.